# Aerobic Exercise in the Aquatic Environment Suppresses the Plasma Renin Activity in Individuals with Type 2 Diabetes: A Secondary Analysis of a Randomized Clinical Trial

**DOI:** 10.3390/ijerph21070938

**Published:** 2024-07-18

**Authors:** Rodrigo Sudatti Delevatti, Larissa dos Santos Leonel, João Gabriel da Silveira Rodrigues, Ana Carolina Kanitz, Cristine Lima Alberton, Gisele Agustini Lovatel, Ionara Rodrigues Siqueira, Luiz Fernando Martins Kruel

**Affiliations:** 1Department of Physical Education, Sports Center, Federal University of Santa Catarina, University Campus Trindade, Florianópolis 88040-900, SC, Brazil; leonellari@gmail.com (L.d.S.L.); gilovatel@gmail.com (G.A.L.); 2Department of Physical Education, School of Physical Education, Physiotherapy and Occupational Therapy, Federal University of Minas Gerais, University Campus, Pampulha, Belo Horizonte 31310-25, MG, Brazil; joaogabrielsrod@gmail.com; 3Department of Physical Education, Federal University of Rio Grande do Sul, Porto Alegre 90040-060, RS, Brazil; ana_kanitz@yahoo.com.br (A.C.K.); luiz.kruel@ufrgs.br (L.F.M.K.); 4Department of Physical Education, Federal University of Pelotas, Pelotas 96020-220, RS, Brazil; tinialberton@yahoo.com.br; 5Department of Pharmacology, Institute of Basic Health Sciences, Federal University of Rio Grande do Sul, Porto Alegre 90040-060, RS, Brazil; ionara@ufrgs.br

**Keywords:** type 2 diabetes, renin-angiotensin system, aquatic exercise, cardiovascular exercise

## Abstract

To compare the acute effects of aquatic walking/running versus dry-land walking/running on blood glucose and plasma renin activity (PRA) in individuals with type 2 diabetes, participants with type 2 diabetes performed deep-water or dry-land walking and/or running sessions in a swimming pool or on an athletics track, respectively. Both sessions comprised seven blocks of 3 min at 85–90% of the heart rate deflection point (HRDP), interspersed with 2 min at <85% HRDP, totaling 35 min, with a 48 h interval between sessions. PRA and blood glucose were assessed before and immediately after the sessions. Generalized estimation equations were used to verify the session effects, with the Bonferroni post hoc test, considering the significance level as 0.05. Twelve individuals (53.2 ± 8.9 years) diagnosed with type 2 diabetes for 6.3 ± 6.34 years participated in the study. A reduction in PRA was found only after the aquatic session (−7.75 ng/mL/h; −69%; *p*: 0.034), while both aquatic and dry-land sessions similarly reduced the blood glucose levels (aquatic: −38 mg/dL, −21%; dry-land: −26 mg/dL, −14%; time effect, *p* = 0.007). Despite yielding similar glycemic reductions as dry-land walking/running, aquatic walking/running led to an expressive decrease in PRA among individuals with type 2 diabetes.

## 1. Introduction

Aerobic exercise is widely recognized as an effective tool for glycemic management [1] and cardiovascular risk factors reduction [2,3] in individuals with type 2 diabetes (T2DM). Walking and running are the most popular forms of aerobic exercise [2,3]. However, the co-occurrence of diabetes with obesity, a condition known as diabesity, and functional disability [4] often hinder the adoption of walking and/or running on dry land, especially for the recommended amount outlined by American Diabetes Association guideline [5].

Alternatively, walking or running in the aquatic environment offers advantages such as reduced musculoskeletal impact [6] and unique physiological alterations [7,8]. Among these physiologic alterations, the suppression of the renin–angiotensin system (RAS) is noteworthy [9], since it is an endocrine system highly associated with blood pressure control [10] and major non-communicable chronic diseases [10,11]. The RAS hyperactivation directly affects insulin secretion, since locally released Ang II impairs the blood perfusion in pancreatic islets, inhibiting the β-cell insulin release [12], and is also related to insulin resistance through the RAS genetic polymorphisms [13]. Therefore, RAS has a close relationship with glycemic control, potentially affecting T2DM management.

A previous study comparing the chronic effects of aquatic- and dry-land-based exercise training in individuals with T2DM have demonstrated similar reductions in HbA1c levels and greater renin–angiotensin suppression in aquatic training, evaluated by plasma renin activity (PRA) and angiotensin II (Ang II) levels [14]. Although chronic effects play a crucial role in guiding clinical treatments, they depend on the cumulative impact and distribution of the numerous sequential exercise sessions. Each session should be prescribed according to several factors, including the acute effect on the target outcome. Similar acute reductions in glycemia were also found between walking/running in aquatic environments versus walking/running on dry-land environments in individuals with T2DM [14,15]. However, to the best of our knowledge, no comparison has been made regarding other clinical outcomes in this population, such as PRA, which is an independent predictor of mortality [16].

Therefore, this study aimed to compare the acute effects of aquatic walking/running versus dry-land walking/running on glycemic and PRA levels in individuals with T2DM. We hypothesized that similar glycemic reductions would occur, whereas a more significant reduction in PRA level would be observed in the aquatic environment.

## 2. Methods

### 2.1. Study Design

This study comprises an ancillary analysis of the clinical trials “Chronic Effects of Two Aerobic-training Models Performed in Water and on Dry Land in Patients With Type 2 Diabetes Mellitus (T2DM)” registered in ClinicalTrials.gov (NCT 01956357). In this trial, the participants allocated to aquatic training underwent aerobic exercise sessions with equal duration and relative intensity to those allocated to dry-land training. PRA and blood glucose were evaluated before and immediately after the sessions. The study was approved by the Research Ethics Committee of Universidade Federal do Rio Grande do Sul (No. 108997).

### 2.2. Participants 

Twelve individuals were recruited through advertisements in newspapers and by revising medical records from the endocrinology sector of the Hospital de Clínicas de Porto Alegre. Eligibility criteria included a T2DM confirmed diagnosis either by laboratory tests or the use of antidiabetic medication. Exclusion criteria were uncontrolled hypertension, severe autonomic or peripheral neuropathy, foot injuries, severe proliferative or non-proliferative diabetic retinopathy, decompensated heart failure, lower limb amputations, chronic renal failure (glomerular filtration rate estimated by MDRD < 30 mL/min/1.73 m^2^) and any muscle or joint problems that could negatively affect the safe performance of the walking and/or running activities. All the participants signed a written consent form. 

### 2.3. Interventions

Participants underwent two walking/running sessions, with the first performed in a dry-land environment, followed by a session in an aquatic environment 48 h later. The aquatic walking/running session was performed in a deep swimming pool, with participants wearing an E.V.A float vest (Floty^®^, Indaiatuba, Brazil) around the waist to avoid contact of the feet with the pool floor. The water temperature was maintained at 32 °C. The dry-land aerobic walking/running session was performed on an athletic track. Both sessions took place at the School of Physical Education of the Universidade Federal do Rio Grande do Sul and were conducted by three exercise professionals with expertise in the field. 

Before the experimental sessions, participants completed three familiarization sessions to familiarize themselves with the exercise techniques and materials. Each session lasted 45 min and consisted of aerobic exercise based on walking and or running, regardless of whether performed in water or on dry land. Each session was divided into three parts: the warm-up, which consisted of 3 min of light-intensity walking; the main part, consisting of walking and/or running lasting 35 min, comprising seven blocks of 3 min at an intensity of 85 to 90% of the heart rate deflection point (HRDP), interspersed with 2 min at an intensity < 85% of HRDP; and the cool-down, consisting of 2 min of light-intensity walking followed by 3 min of standardized stretching, emphasizing the muscle employed in the main part of the session.

Walking/running was the exercise type chosen as it is a form of aerobic modality commonly performed and is more cost-accessible. In addition, the cyclic nature of the modality allows for a more suitable comparison of the effects of the environments (aquatic versus land) of exercise, compared to other modalities.

### 2.4. Assessments for Sample Characterization and Exercise Prescription

Variables, including sex (female and male), age (years), body mass (kg), height (cm), body mass index (BMI; kg/m^2^), duration of diabetes (years), and medication, were assessed to characterize the sample.

A maximum test was carried out on a treadmill to prescribe the aerobic session on dry land. The test started at 3 km/h for 3 min, with 1 km/h increments every 2 min, maintaining a 1% incline. The test was performed until voluntary exhaustion (signaled by hand gestures) and was considered valid according to the recommendations of Howley et al. [17]. 

To prescribe the aerobic session in the aquatic environment, participants performed a maximal running test in a deep pool using the protocol outlined by Kanitz [18]. The test started at an initial cadence of 85 beats per minute for 3 min, followed by increases of 15 beats per minute every 2 min until voluntary exhaustion. The test was stopped when the individual indicated exhaustion or could not maintain the metronome’s pace and the stride amplitude. A 48 h interval was observed between the tests. Individuals were instructed not to eat for 3 h before the tests, to avoid consuming stimulants, and to refrain from engaging in intense physical activity 12 h before the test [19]. The sessions were prescribed using the HRDP, which was determined through visual inspection by two experienced physiologists, considering data from the incremental tests carried out in the aquatic and dry-land environments.

### 2.5. Outcomes Assessment

The outcomes were collected before and immediately after the sessions. Venous blood punction was performed to collect 8 mL of blood in EDTA tubes. Tubes were centrifugated for 10 min at 1500 rpm at −4 °C, and 1 mL of plasma was inserted in dried Eppendorf tubes and stored at −20 °C until analysis. The PRA levels were determined by the radioimmunoassay kit GAMMACOAT (DiaSorin, Macquarie Park, NSW, Australia), while plasmatic blood glucose was measured by the hexokinase method. The details of the flowchart, with the designer of the present study, are shown in Figure 1.

### 2.6. Statistical Analysis

Continuous variables for sample characterization were expressed as the mean and standard deviation, while categorical variables were presented as relative frequency and absolute. Generalized estimation equations (GEE) were used for outcome evaluation (between- and within-session changes), followed by the Bonferroni post hoc tests. Outcome data were expressed as mean and standard error, with a significance level set at 5%. All analyses were performed using IBM SPSS version 21.0 (IBM Corp., Armonk, NY, USA).

## 3. Results

Twelve individuals of both sexes, ranging in age from 37 to 63 years, participated in this study. All participants were classified as obese (66% with grade I obesity, 25% grade II, and 8% grade III), and 90% of the sample exhibited systemic arterial hypertension. The detailed characteristics of the sample are presented in Table 1. 

Experimental sessions were well tolerated by all individuals, without any adverse event. Regarding PRA, interaction time*session was found (*p* = 0.04). The post hoc analysis demonstrated a PRA reduction only after the aquatic session (−7.75 ng/mL/h; −69%; *p*: 0.034) (Figure 2). 

Blood glucose showed a time effect (*p* = 0.007), with reductions observed in both sessions (aquatic: −38 mg/dL, −21%; dry-land: −26 mg/dL, −14%) (Figure 3).

## 4. Discussion

The present study showed that a single session of aquatic aerobic exercise, based on walking and/or running, effectively reduced the PRA, whereas the dry-land session did not yield the same effect. Furthermore, both aquatic and dry-land aerobic exercise sessions exhibited similar reductions in blood glucose levels among individuals with T2DM.

Unprecedentedly, our results highlight the potential effect of aquatic aerobic exercise on reducing PRA. Incipient studies showed that a single bout of aerobic exercise can modulate the circulating levels of some RAS peptides in young males without diseases [20]. The RAS, a physiological system that regulates blood pressure and fluid balance [21], is hyperactive in individuals with T2DM [11]. PRA serves as a marker of RAS activity and may reflect cardiovascular system health [16], with abnormally elevated levels in both hypertensive and normotensive people with T2DM [22]. Prospective studies showed a higher risk of major vascular events (1.38 hazard ratio) and cardiovascular death (1.89 hazard ratio) in the highest fifth of PRA compared to the lowest fifth of PRA in subjects older than 55 years with diabetes [16]. Additionally, exercise interventions have demonstrated the potential to reduce PRA, with studies reporting decreases in PRA following 4 to 37 weeks of exercise in healthy populations [23]. 

In this context, our previous research showed that 12 weeks of aquatic or dry-land exercise training reduced PRA and Ang II in subjects with T2DM [14]. This study found a superior effect in RAS markers in water, particularly for PRA levels. The per-protocol analysis demonstrated a reduction only for the aquatic exercise group. In contrast, the intention-to-treat analysis demonstrated reductions in both groups, albeit with a higher magnitude for those training in the aquatic environment (67% vs. 47% for dry-land exercise). In line with the current study’s findings, a single aquatic exercise session exhibited superior effects in decreasing PRA levels, and the cumulative effect of acute exercise sessions (namely, exercise training) may potentiate the PRA reduction. The significant effect on PRA in the aquatic environment compared to the dry-land environment can be attributed to physiological changes resulting from immersion. The hydrostatic pressure and combined with the facilitated heat exchange by conduction and convection lead to central hypervolemia with hemodilution, and a reduction in osmotic pressure, which triggers a neuroendocrine cascade with the consequent suppression of the RAS [9,24,25].

Concerning blood glucose levels, aquatic and dry-land walking/running effectively reduced blood glucose. Acute reductions after aerobic training sessions are widely reported in the literature [15,26,27,28,29]. However, the studies are often conducted in a dry-land environment and rarely compare different training environments. For instance, Delevatti et al. [15] compared the acute effect of three different sessions (the first of each mesocycle) of a 9-week (3 mesocycles of 3 weeks) aerobic training program performed in aquatic versus dry-land environments, showing similar reductions in both groups throughout the mesocycles.

Although the cross-talk between insulin resistance and RAS markers has been widely discussed, demonstrating that the suppression of the RAS axis is directly linked to changes in the glycemic system with repercussions on diabetes control [30,31] and the prevention of T2DM [30,32,33], this interaction between RAS and exercise glycemic control is limited. Our findings currently demonstrate that the aquatic environment provides acute suppression in the PRA; whereas a previous study by the same author [15] suggests that the cumulative effect of sessions optimizes chronic reductions in the aquatic environment, this marker seems to be more quickly affected by the benefits of immersion. Also, various factors influence acute and chronic glycemic responses to exercise, such as intensity and/or volume [15,34], with no evidence of differences between aquatic or dry-land training environments, both acutely, by effects on acute glycemic behavior [15] and chronically, measured by HbA1c [14,35]. Nevertheless, since the literature is incipient on aquatic versus dry-land exercise in the control of T2DM, other mechanisms may be influenced by the aquatic environment, which could potentiate cardiometabolic improvements in studies with a more extended intervention period.However, the evidence from this cross-talk makes us speculate that, from a long-term perspective, the considerable RAS suppression in the aquatic environment may favorably modulate the glycemic status of patients with T2DM. Further research is needed to elucidate the interaction of the exercise environments (aquatic and dry-land), especially regarding the cumulative effect of training sessions. This additional research will clarify whether acute exercise impacts important but not commonly measured markers, such as PRA, and whether it may provide better chronic neuroendocrine modulation, thereby optimizing T2DM treatment. 

It is important briefly discuss the findings based on the design adopted. Cross-over trial designs have been utilized for evaluating the acute effects of different exercise sessions. This design model excludes within-subject variability, but it requires a sufficient time between sessions for washout occurrence. In this study, we determined a 48 h interval as adequate for the evaluated outcomes in response to a modest dose (duration and intensity) in different environments.

This study’s strengths are that exercise is prescribed according to the anaerobic threshold using the HRDP method, a low-cost and easy-to-apply approach. In addition, maximal tests were carried out in aquatic and dry-land environments, ensuring the physiological responses in each specific environment and providing equivalent prescriptions between them. Another notable point is the comparison between training environments, offering insights into acute exercise responses in mechanistically and clinically relevant markers in T2DM. However, the absence of land and aquatic control sessions without exercise, as well as dietary control and the non-randomization of the order of the sessions, can be recognized as potential limitations. 

## 5. Conclusions

In individuals with type 2 diabetes, aquatic walking/running acutely reduces plasma renin levels to a greater extent than dry-land walking/running, while eliciting similar reductions in the glycemic response between environments.

## Figures and Tables

**Figure 1 ijerph-21-00938-f001:**
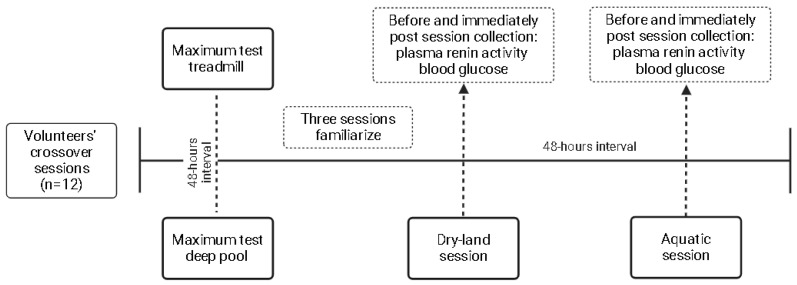
Study design.

**Figure 2 ijerph-21-00938-f002:**
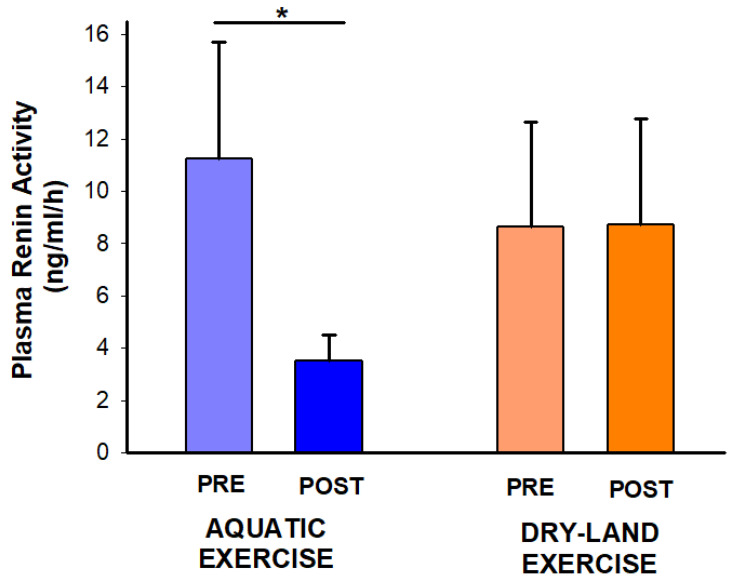
Effect of two exercise sessions on plasma renin activity. The asterisk (*) denotes a significant within-session comparison at *p* < 0.05.

**Figure 3 ijerph-21-00938-f003:**
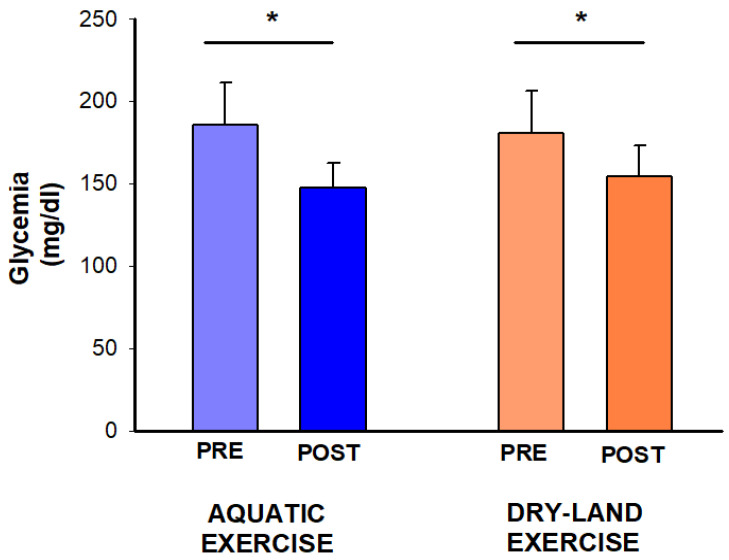
Effect of two exercise sessions on blood glucose. The asterisk (*) denotes a significant within-session comparison at *p* < 0.05.

**Table 1 ijerph-21-00938-t001:** Sample characterization (n = 12).

Variables	$\bar{x}$ ± sd/n
Age (years)	53.2 ± 8.9
Sex (F/M)	6/6
Body Mass (kg)	95.5 ± 14.0
Height (cm)	167.7 ± 11.0
BMI (kg/m^2^)	34.0 ± 4.0
Duration of diabetes (years)	6.3 ± 6.3
**Medical treatment**	**n (%)**
Metformin	11 (91%)
Sulphonylurea	6 (50%)
Insulin	5 (41%)
ACE inhibitors	5 (41%)
ARAs	6 (50%)
Alpha-2 Adrenergic Receptor Agonists	1 (8%)
Dihydropyridine calcium channel blockers	1 (8%)
Diuretics	7 (58%)
Beta blockers	3 (25%)
Statins	8 (66%)
Acetylsalicylic acid	6 (50%)
Proton pump inhibitors	1 (8%)
Antiepileptics	1 (8%)
Benzodiazepines	2 (16%)
Antidepressants	2 (16%)

ACE: angiotensin-converting enzyme; ARAs: angiotensin receptor antagonist; BMI: body mass index; cm: centimeters; F: female; M: male. Continuous data are presented as mean and standard deviation ($\bar{x}$ ± sd); Categorical data are presented in absolute frequency (n sample) and relative frequency (%).

## Data Availability

Data can be requested from the authors.

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
