# Peer review of "Aerobic Exercise in the Aquatic Environment Suppresses the Plasma Renin Activity in Individuals with Type 2 Diabetes: A Secondary Analysis of a Randomized Clinical Trial"

_ijerph, 2024, doi:10.3390/ijerph21070938_

Round 1

Reviewer 1 Report

Comments and Suggestions for Authors

In the present study, authors demonstrated acute effect of aquatic walking and running versus dry-land walking and running and the association with blood glucose and plasma renin activity in patients type 2 diabetes mellitus. The study has interesting findings from patients with T2DM and portrayed significance results using various statistical analysis. I have listed few suggestions that can be addressed in the revised manuscript by the authors before considering the manuscript further.

1.    Please mention guidelines as American Diabetes Association (ADA) in the page no. 1.

2.    Please include number of participants and also exclusion criteria if any in the method section “2.2. Participants”.

3.    I suggest authors to change title of the manuscript as “A pilot randomized clinical trial” since you had very less participants (only 12 participants) in the present study.

4.    I would like to suggest authors to include cohorts’ characteristics and no of participants as a flow chart in the manuscript, which would give readers more clarity on the present study.

5.    Please mention method of detection for glucose in the method section with reference since you have used capillary blood glucose in the study.

6.    You have mentioned SRA markers in the page no. 6 of 8. What is SRA markers? Give expansion.

7.    Please check typographical and grammatical errors throughout the manuscript.

Author Response

General comments

In the present study, authors demonstrated acute effect of aquatic walking and running versus dry-land walking and running and the association with blood glucose and plasma renin activity in patients type 2 diabetes mellitus. The study has interesting findings from patients with T2DM and portrayed significance results using various statistical analysis. I have listed few suggestions that can be addressed in the revised manuscript by the authors before considering the manuscript further. 

Comment 1: Please mention guidelines as American Diabetes Association (ADA) in the page no. 1.

Response 1: The mention was inserted.

Comment 2: Please include number of participants and also exclusion criteria if any in the method section “2.2. Participants”.

Response 2: The required changes were performed in the original text.

Comment 3: I suggest authors to change title of the manuscript as “A pilot randomized clinical trial” since you had very less participants (only 12 participants) in the present study.

Response 3: I acknowledge that according to other methodological frameworks, especially traditional parallel studies, our current sample size might be deemed preliminary. However, the crossover model utilized in this study presents an effective alternative aimed at reducing variability among participants. By employing each participant as their own control, this approach not only mitigates the impact of external confounding factors but also enhances the precision in detecting differences between treatments. Furthermore, existing research in this field employing similar methodological models supports the practice of smaller sample sizes in crossover studies, thereby ensuring the attainment of robust and dependable results.

References:

Mendes R, Sousa N, Themudo-Barata JL, Reis VM. High-Intensity Interval Training Versus Moderate-Intensity Continuous Training in Middle-Aged and Older Patients with Type 2 Diabetes: A Randomized Controlled Crossover Trial of the Acute Effects of Treadmill Walking on Glycemic Control. Int J Environ Res Public Health. 2019 Oct 28;16(21):4163. doi: 10.3390/ijerph16214163. PMID: 31661946; PMCID: PMC6862460.

Moreno-Cabañas A, Morales-Palomo F, Alvarez-Jimenez L, Mora-Gonzalez D, Ortega JF, Mora-Rodriguez R. Metformin and exercise effects on postprandial insulin sensitivity and glucose kinetics in pre-diabetic and diabetic adults. Am J Physiol Endocrinol Metab. 2023 Oct 1;325(4):E310-E324. doi: 10.1152/ajpendo.00118.2023. Epub 2023 Aug 16. PMID: 37584610.

Eshghi SR, Fletcher K, Myette-Côté É, Durrer C, Gabr RQ, Little JP, Senior P, Steinback C, Davenport MH, Bell GJ, Brocks DR, Boulé NG. Glycemic and Metabolic Effects of Two Long Bouts of Moderate-Intensity Exercise in Men with Normal Glucose Tolerance or Type 2 Diabetes. Front Endocrinol (Lausanne). 2017 Jul 11;8:154. doi: 10.3389/fendo.2017.00154. PMID: 28744255; PMCID: PMC5504214.

Comment 4: I would like to suggest authors to include cohorts’ characteristics and no of participants as a flow chart in the manuscript, which would give readers more clarity on the present study.

Response 4: The flowchart was included.

Comment 5: Please mention method of detection for glucose in the method section with reference since you have used capillary blood glucose in the study.

Response 5: The method of detection of plasma glucose was the hexokinase method, not capillary blood glucose. This information was previously inserted in the original text, see in ‘2.5 – Outcomes assessment’.

Comment 6: You have mentioned SRA markers in the page no. 6 of 8. What is SRA markers? Give expansion.

Response 6: The correct acronym is RAS. The description was inserted in the second paragraph of the ‘Introduction section’ and refers to Renin-angiotensin system.

Comment 7: Please check typographical and grammatical errors throughout the manuscript.

Response 7: A typographical and grammatical check was performed.

Reviewer 2 Report

Comments and Suggestions for Authors

The work presented by the authors has relative relevance for the scientific and clinical community. I find the pursued objectives of interest, but I observe shortcomings in the study's design. It does not describe exactly how the different tests were conducted, whether they were performed in all subjects in the same order or if they were randomized. Additionally, there are gaps in the recruitment of subjects, as they should be a more heterogeneous population. Furthermore, the study was conducted on only 12 subjects, which is an excessively small sample size to extrapolate the results, particularly in relation to diabetes. Therefore, despite the work arousing some interest, I believe it cannot be proposed for publication.

Author Response

Comments: The work presented by the authors has relative relevance for the scientific and clinical community. I find the pursued objectives of interest, but I observe shortcomings in the study's design. It does not describe exactly how the different tests were conducted, whether they were performed in all subjects in the same order or if they were randomized. Additionally, there are gaps in the recruitment of subjects, as they should be a more heterogeneous population. Furthermore, the study was conducted on only 12 subjects, which is an excessively small sample size to extrapolate the results, particularly in relation to diabetes. Therefore, despite the work arousing some interest, I believe it cannot be proposed for publication.

Responses: Dear reviewer, we appreciate your feedback, and in order to enhance the exercise prescription method, several changes have been made to section “2.4 - Assessments for sample characterization and exercise prescription”.  We understand your concern regarding the sample size of 12 participants; however, similar crossover studies involving patients with type 2 diabetes and physical exercise also demonstrate comparable sample sizes. The crossover model is known to minimize the impact of external confounding variables and allows for more accurate detection of differences between treatments, as participants act as their own controls. Furthermore, to address the inherent aspects raised in the chronic clinical trial study, recruited participants met eligibility criteria for safe training. We included individuals of both sexes using various hypoglycemic medications that represent different disease control mechanisms, as well as some variation in disease diagnosis time, factors reflecting certain heterogeneity among the patients themselves. We chose to exercise caution in interpreting and generalizing the results, acknowledging the non-randomized order as one of the study's limitations.

Reviewer 3 Report

Comments and Suggestions for Authors

To authors,

The topic “Aerobic exercise in the aquatic environment suppresses the plasma renin activity in individuals with type 2 diabetes: a secondary analysis of a randomized clinical trial.” 

This manuscript is an interesting and novelty moderate of study in the field. However, before acceptance for publication, some points of the manuscript need to be revised and made clear for correct understanding.

Minor revision is required.

The comments are following:

The topic “Aerobic exercise in the aquatic environment suppresses the plasma renin activity in individuals with type 2 diabetes: a secondary analysis of a randomized clinical trial.” 

This manuscript is an interesting and novelty moderate of study in the field. However, before accepting for publication, some point of manuscript needs to revise and make it clear for correct understanding.

Minor revision is required.

1. The introduction part should be more highlighted in the significant point or pain point that why the current work needs to determine.

2. The discussion section, there is no relative of the results between of each figure. Authors try to correct the vital and compare or summarize with the outstanding point from the finding.

3. What is the reason behind the choice of selected intervention of the participant with working/running for the study?

4. How the authors try to random the sample to avoid any variation of clinical test

5. The authors should include the source of chemicals used in the present investigation in the materials and methods section.

Best regards,

Author Response

General comment: This manuscript is an interesting and novelty moderate of study in the field. However, before accepting for publication, some point of manuscript needs to revise and make it clear for correct understanding.

Minor revision is required.

 The comments are following:

Comment 1: The introduction part should be more highlighted in the significant point or pain point that why the current work needs to determine.

Response 1: To highlight the research problem, we have adjusted the introduction.

Comment 2: The discussion section, there is no relative of the results between of each figure. Authors try to correct the vital and compare or summarize with the outstanding point from the finding.

Response 2: We rewrite some points of the discussion to clarify the explanations of the findings.

Comment 3: What is the reason behind the choice of selected intervention of the participant with working/running for the study?

Response 3: Walking/running was chosen mainly due two reasons, that follow:

  1. It is the more common and cost-accessible aerobic activity.
  2. By the cyclic nature of the activity the comparison about effects of environments (aquatic versus land) is more suitable. Other aerobic modalities as cycling, hydrogymnastic or resistance training have substantial differences in prescriptions for land and aquatic environment, making difficult the comparison.

We inserted, of briefly form, the reason of choice in methods.

Comment 4: How the authors try to random the sample to avoid any variation of clinical test.

Response 4: This study has a crossover design and was conducted in initial phase of a major randomized clinical trial in which 35 patients were randomly allocated to aquatic or land training groups. For the secondary analysis, with more mechanistic and acute effect interest, we used only patients allocated in aquatic group because they had exercise tests for prescription in two environments, facilitating logistics. Ideally, the order of sessions should be randomized, by this was not possible, by conduction of the main trial. The non-randomized order of session was inserted as limitation.  

Comment 5: The authors should include the source of chemicals used in the present investigation in the materials and methods section.

Response 5: We don't understand the questioning. But in order to try to clarify, in the methods in topic “2.5. Outcomes assessment” we have improved the description of the blood glucose and plasma renin activity collections. If the question relates to the drugs used by the participants, their medication was noted according to their doctor's prescription during the recruitment process, and no changes in dosage were advised.

Round 2

Reviewer 2 Report

Comments and Suggestions for Authors

Espite the fact that the authors addressed revisions and made improvements in the presented work, I still believe it could be relevant to the scientific and clinical community. However, methodological deficiencies observed in the first version of the paper still remain.

Author Response

Dear reviewer, unfortunately we were unable to adjust methodological issues such as the sample size. However, we tried to greatly improve the study description, including another paragraph in the discussion talking about the cross-over design in the scenario of the present study. In any case, we appreciate the review carried out and the opportunity to improve the manuscript and review future designs.